# Asymptomatic Aortic Stenosis in an Older Patient: How the Geriatric Approach Can Make a Difference

**DOI:** 10.3390/diagnostics13050909

**Published:** 2023-02-27

**Authors:** Alberto Finazzi, Adriana Antonella Bruni, Stefano Nistri, Giuseppe Bellelli

**Affiliations:** 1School of Medicine and Surgery, University of Milano-Bicocca, 20900 Monza, Monza-Brianza, Italy; 2Acute Geriatric Unit, S. Gerardo Hospital, 20900 Monza, Monza-Brianza, Italy; 3Cardiology Service C.M.S.R. Veneto Medica, Altavilla Vicentina, 36077 Vicenza, Italy

**Keywords:** aortic stenosis, elderly, comprehensive geriatric assessment, transcatheter aortic valve implantation, heart team

## Abstract

We present a case report of an older patient with aortic stenosis who was managed before and after transcatheter aortic valve implantation by a team of cardiologists but without the support of a geriatrician. We first describe the patient’s post-interventional complications from a geriatric perspective and afterwards, discuss the unique approach that the geriatrician would have provided. This case report was written by a group of geriatricians working in an acute hospital, along with a clinical cardiologist who is an expert in aortic stenosis. We discuss the implications for modifying conventional practice in tandem with existing literature.

## 1. Introduction

Although aortic stenosis (AS) patients are predominantly elderly, geriatricians seldom are involved in the decision-making processes of the Heart Team. Moreover, a comprehensive geriatric assessment (CGA) is a key factor contributing to the optimization of care for geriatric patients but is rarely performed. In this clinical case, we discuss the importance of involving a geriatrician in the decision-making processes of the heart-team and highlight the clinical implications of using the CGA to identify the complex needs of older patients with AS. A clinical cardiologist with specific competence in evaluating AS aided us in the discussion of the case report.

## 2. Case Presentation

An 81-year-old man was referred to our hospital by his general practitioner (GP) for a comprehensive cardiological evaluation due to a severe systolic murmur in the absence of dyspnea, syncope, and angina. The patient weighed 60 kg and was 172 cm tall with a body mass index (BMI) of 20.3 kg/m^2^ and body surface area (BSA) 1.69 m^2^. He smoked tobacco for roughly 20 years (1 pk/day), and stopped 30 years ago. He had a past medical history of systemic arterial hypertension, hypercholesterolemia, stage 2 chronic renal disease with an estimated glomerular filtration rate of 60 mL/min, anxiety, and depression. Additionally, his daughter reported mild cognitive disorder, which had not been further investigated. Medications included atorvastatin 40 mg once a day orally, enalapril/hydrochlorothiazide 20 mg + 12.5 mg once a day orally, paroxetine 20 mg once a day orally, and alprazolam 0.50 mg once a day orally (before sleep).

The electrocardiogram showed sinus rhythm and was otherwise unremarkable. Blood pressure was 130/80 mmHg. Echocardiography showed normal left ventricular mass (108 g/m^2^) while ejection fraction was 49%, and there was a grade I diastolic dysfunction of the left ventricle. The aortic valve was severely calcified, and hypo mobile with a peak valvular velocity of 4.2 m/s and a mean gradient of 43 mmHg. The aortic valve area was 0.9 cm^2^ (0.52 cm^2^/m^2^). Tricuspid regurgitation was mild with a peak velocity of 2.9 m/s, and the right ventricle was normally sized with tricuspid annular plane systolic excursion of 19 mm, E/e’ was 15. Consistently, the patient was diagnosed as having asymptomatic high-gradient AS with reduced ejection fraction and referred for evaluation for aortic valve replacement.

In view of the age older than 75 years, eligibility to transcatheter aortic valve implantation (TAVI) was assessed, as indicated by the 2021 ESC/EACTS guidelines for the management of valvular heart disease. Blood test showed hemoglobin 12.8 g/dL, hematocrit 35%, creatinine 0.8 mg/dL, and N-terminal pro-B-type natriuretic peptide 411 pg/mL (NT-proBNP, normal value < 125 pg/mL). The operative risk was low by the Society of Thoracic Surgeons score (2.112%) and by the European System for Cardiac Operative Risk Evaluation II score (4.74%). To assess the eligibility to a transcatheter aortic valve implantation (TAVI), the patient was then evaluated with chest and abdomen computed tomography that showed no significant stenosis of the coronary arteries and confirmed the feasibility of this approach. Therefore, the Cardiology team decided to treat the patient with a balloon-expandable Edwards Sapien 3 via transfemoral access under procedural sedation and analgesia. The procedure was performed successfully 2 months after the referral, with no aortic regurgitation at standardized aortography and by transthoracic echocardiography one day after valve implantation.

On the first day after the procedure, while staying in the ICU, the patient developed mild restlessness and reduced awareness of the environment. He was not oriented in place and time. Blood examinations found hyponatremia (Na 131 mmol/L) and a slight increase in C-reactive protein serum levels (CRP 5 mg/dL), without an increase of white cells. At night, agitation became severe, the patient tried to get out of bed, and self-removed both the venous lines and the bladder catheter, provoking hematuria. The cardiologist on call decided to replace the bladder catheter and ordered promazine 50 mg intramuscularly with little benefit to the patient’s restlessness.

From day III to day V, there was a close-up fluctuation from drowsiness to hyperactivity, which was treated with promazine. To improve the patient’s behavior, the cardiologist decided to remove the bladder catheter, but acute urinary retention developed, which compelled them to maintain it in situ until discharge. On the sixth day, the patient was discharged from hospital in good clinical status, with a scheduled cardiological checkup in one month and the suggestion to contact a nurse for the management of bladder catheter.

Eight days after the discharge, the patient was referred by his GP to the emergency department of the same hospital due to persistent agitation, fever, and genitourinary pain. Biochemical examinations showed leukocytosis (12.30 × 10^3^/mL, neutrophils 9.59 × 10^3^/mL), further increased CRP (8.71 mg/dL), thrombocytosis (platelet = 450 × 10^3^/mL), and hyponatremia (Na 130 mmol/L), in addition, NT-proBNP 1223 pg/mL and troponin T 13 ng/L. The electrocardiogram, the bedside echocardiogram, and the chest X-ray showed nothing significant.

The patient was then transferred to the Acute Geriatric Unit (AGU) with the diagnosis of suspected urinary tract infection (UTI), which was then confirmed at the urine culture (*E. coli*). The 4AT (Figure 1) [1] scored 10/12, suggesting the presence of delirium, which was then confirmed according to the Diagnostic and Statistical Manual of Mental Disorders (DSM-5) criteria [2]. The causes of delirium were treated by withdrawing sedatives (alprazolam, promazine), correcting electrolyte disorders by infusion of saline solution, treating UTI with antibiotic (ceftazidime 2 g intravenous twice a day for a week), and removing bladder catheter. Furthermore, to prevent acute urinary retention, extemporaneous catheterizations were scheduled during the day. The presence of a family member was allowed at the bedside to improve the patient’s well-being, and an occupational therapist was activated to improve the patient’s activities of daily living, as the literature suggests efficacy of these approaches to treat delirium [3]. Low-dose trazodone (25 mg mid-afternoon and 50 mg after dinner) and slow-release melatonin (2 mg after dinner) per os were started, to improve circadian rhythm and sleep. The patient was then discharged toward a rehabilitation unit after a total AGU length of stay of 15 days.

## 3. Discussion

### 3.1. The Added Value of the Comprehensive Geriatric Assessment and the Geriatric Approach

The global population of individuals aged 65 years and over is growing at an unprecedented rate, and it is expected that this trend will continue or even increase in the future [4]. However, not all individuals age in the same way, as there is significant heterogeneity in life trajectories between persons of the same age [4,5], with substantial evidence that chronological age is an inaccurate marker both of aging and of clinical outcomes. Moreover, there is an urgent need to implement tools and methods to capture the heterogeneity of aging and, thus, the biological complexity of the individuals in order to tailor treatment and interventions.

The CGA is a “multidimensional interdisciplinary diagnostic process focusing on multiple health problems of an old person, in order to develop a coordinated and integrated plan for treatment and long term follow up” [6]. Unlike standard medical evaluation, CGA also assesses non-medical domains, including cognitive, functional, nutritional, and socio-environmental status, and it is considered the best approach in geriatric medicine for identifying the biological complexity of an older adult [7].

Frailty is the paradigmatic construct of the clinical and biological complexity of an older person. It is defined as a medical syndrome with multiple causes and contributors that is characterized by diminished strength, endurance, and reduced physiologic function that increases an individual’s vulnerability to developing increased dependency and/or death when exposed to a stressor [8]. The assessment of frailty in older adults is key in the geriatric approach since it may enable physicians to predict adverse events, including complications after procedures, functional and cognitive decline, falls, disability, and mortality [9].

In the presented case, the CGA enabled us to capture a clinical picture of the patient’s complexity that was overlooked both by the GP and cardiologists (Table 1). The patient was widowed and lived at home alone with a daughter, who could only support him for a few hours during the day. This suggested that the social support was poor. Although he was reported to be independent in all the basic activities of daily living, he needed help in shopping, housekeeping, using transport, handling money, and preparing food. Moreover, he had severe nutritional problems despite an apparently normal BMI, with calf circumference measuring 29.5 cm (which is below the normal reference values), serum albumin levels of 3.1 g/dL (normal value above 4.0 g/dL), and the Mini Nutritional Assessment (MNA) scoring 8/14, suggesting overt malnutrition. Overall, the Clinical Frailty Scale (CFS) scored 5, and a 50-item electronic Frailty Index (FI) was 0.34, suggesting a moderate to severe frailty level.

The cognitive assessment on admission to the AGU revealed the presence of delirium superimposed on mild dementia. Delirium is a geriatric syndrome characterized by acute impaired attention and awareness, fluctuating course, and global cognitive dysfunction [2]. It is particularly common among older and critically ill patients and frequently arises as a complication of surgical or interventional procedures, acute medical conditions, intoxication with or withdrawal of medication, or electrolyte or metabolic imbalances [10,11,12].

Both TAVI and frailty are significantly associated with the occurrence of delirium [13,14]. Importantly, delirium can be prevented using non-pharmacological approaches in individuals at risk. A recent systematic review and meta-analysis of 44 articles identified 14 high-quality trials, finding that a bundle of non-pharmacological interventions (including limited use of psychoactive drugs, reorientation, promotion of sleep, maintenance of adequate hydration and nutrition, early mobilization, and provision of visual and hearing adaptations) was able to significantly decrease delirium incidence (odds ratio [OR], 0.47; 95% CI, 0.37–0.59) and risk of fall (OR 0.58; 95% CI, 0.35–0.95) [15]. In the described case, the detection of delirium would have prompted physicians to reconcile drugs and withdraw those unnecessary or potentially harmful. For instance, the patient was prescribed alprazolam, a benzodiazepine, which is not recommended at a first line for sleep or behavioral disorders in older people [16].

Once delirium occurs, a systematic search of all its potential causes should begin immediately. Post-surgical delirium is generally triggered by procedures and pain, with cytokines and other mediators of inflammation that enter the brain parenchyma (through the blood-brain barrier) and activate both the microglia and astrocytes, leading to transient or persistent neuroinflammation of the brain [17]. Other suspected pathophysiological mechanisms include impaired glucose supply at mitochondria in the brain, neuroendocrine disorders, and sleep-wake cycle disruption [17].

However, the use of bladder catheters, venous lines, nasogastric tubes, and other devices, as well as inappropriate medications (Table 2), can exacerbate or trigger delirium [17]. From a clinician’s perspective, antipsychotics (such as promazine) should be avoided in patients with delirium because they’re ineffective and potentially dangerous [16,18]. If agitation is substantial, with severe patient distress, a drug with little anticholinergic burden (such as haloperidol, Table 2) should be preferred and used for the shortest possible time instead of promazine [18]. Furthermore, bladder catheters and venous lines should be removed as soon as possible, and early mobilization promoted [3].

Unfortunately, despite clear evidence to support their use, CGA is not routinely performed (and thus frailty is not evaluated), and non-pharmacological approaches are underused in hospital wards [19,20]. It is, therefore, urgently required to switch the usual pattern of care for older people towards a more active interaction between specialists, with geriatricians being the case managers of older and complex patients.

### 3.2. Considerations from a Clinical Cardiologist

Degenerative AS is a common, age-related condition whose prevalence and incidence are often underestimated. There is an increased risk of mortality across the whole spectrum of AS, independently from many comorbidities and treatments [21,22]. Overall, AS confers an increased risk of mortality or adverse cardiovascular events, such as myocardial infarction, heart failure, and stroke. Moreover, the rate of hemodynamic progression of the disease is unpredictably rapid in a substantial proportion of individuals, negatively affecting their outcomes [23,24].

The progressive introduction of multiple innovative technologies has dramatically impacted the scenario of aortic valve replacement (AVR) with a disruptive role of transcatheter aortic valve replacement in expanding the overall numbers, risk profile, and outcome of patients undergoing AVR [25,26].

TAVI is usually successful in older patients, as shown by several studies [27]. However, a non-marginal share of them experiences an overall poor outcome, which is not related to the procedure itself but largely attributable to frailty [28,29,30]. These observations prompt the need for the clinical cardiologist to acknowledge that frailty is a major determinant of the patient’s functional recovery and outcomes. Therefore, ideally, the presence of frailty should be systematically investigated in older patients before AVR, and its management should be considered as a treatment target to be pursued in parallel to surgery. In the context of evaluation to AVR, frailty is frequently considered a reason for choosing conservative medical approaches [31,32]. This reflects a misconception that should be challenged by emphasizing that frailty should not be equated with futility [31].

The reported case offers multiple points for discussion. First, it underscores that AS is underdiagnosed and disregarded. Second, in an older patient, a combination of comorbidities and dependence in daily activities may often mask the presentation of the typical symptoms of AS, implying that AS is often diagnosed at an advanced stage [33,34,35]. Even more importantly, this case report clearly shows that an accurate evaluation of the cardiac disease is not mirrored by a similarly accurate evaluation of the patient. Thus, our patient’s clinical course has been dramatically characterized by several situations that a proper pre-procedural geriatric assessment could have detected, resulting in an appropriate strategy aimed at de-frailing on a short-term basis using a pragmatic multicomponent intervention [36]. Such an approach should be integrated by an early exercise intervention with potential positive impacts both on clinical outcomes and the ability to maintain independent living [37].

To date, the patient journey of individuals referred to AVR, mostly consists of a series of tests aimed at assessing appropriateness, choice of procedure for safety and effectiveness, and their optimization to avoid complications, or detecting/treating them appropriately, and to reduce as much as possible the in-hospital stay. The time has come to integrate this process with a brief screening test to identify the patients who may require further CGA. The Essential Frailty Toolset may be particularly appropriate to this aim [38]. Based on the results of the latter, a multicomponent (e.g., exercise, nutritional, cognitive) comprehensive, home-based approach should be offered during the pre-hospitalization phase to de-frail older patients with frailty. Moreover, during the in hospital stay, an appropriate strategy based on prompt detection and appropriate treatment of delirium and early exercise training, environmental, cognitive, and pharmacological interventions should be performed. Ideally, a personalized post-hospital rehabilitation program should also be considered [39].

Available evidence suggests the need for the actual profile of all centers performing aortic valve interventions should be combined with an integrated geriatric approach moving from a traditional bedrest-based hospitalization—possibly resulting in sarcopenia [40]—to one that considers care of cognitive function and functional capacity as integral components of any cardiovascular procedure in older adults.

## 4. Conclusions

Based on this case record and on the existing literature, we recommend the involvement of the geriatricians both in the decision-making processes of the heart team and in the co-management of the periprocedural period for older patients with AS. In particular, the use of the CGA by a geriatric team is crucial to provide optimal care and identify the complex needs of these patients before and after the surgical procedures.

## Figures and Tables

**Figure 1 diagnostics-13-00909-f001:**
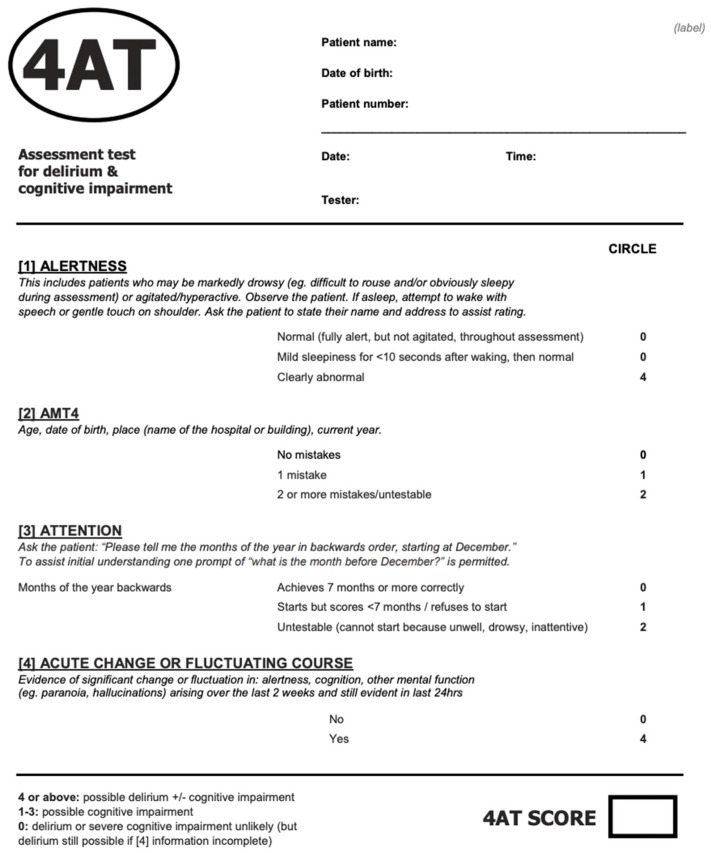
The 4AT scale for the screening of delirium. This scale contains a few simple questions that guide the clinician to define the presence of delirium (for further information, www.the4at.com, accessed on 30 December 2022).

**Table 1 diagnostics-13-00909-t001:** The CGA of the clinical case. We suggest paying great attention to social support: A poor social network can lead to an overestimation of the real patient’s independence. Moreover, BMI is not sufficient to assess nutritional status.

Domain	Items to Be Assessed	Findings
Social circumstances	Informal support from family or friends	He lived at home alone, daughter able to support her father only for a few hours during the day
Medical	Co-morbid conditions and disease severity Nutritional Status	BMI 20.3 kg/m^2^ BSA 1.69 m^2^ Calf circumference 29.5 cm Albumin levels 3.1 g/dL MNA 8/14
Mental health	Cognition Mood and anxiety	MMSE 23/30 Anxiety and depression
Functional capacity	Basic activities of daily living Instrumental activities of daily living	BADL 6/6 IADL 3/8
Medication review	Polypharmacy Anticholinergic Burden	Number of active ingredient 5

Legend: Mini Nutritional Assessment (MNA), Mini-Mental State Examination (MMSE), Basic Activities of Daily Living (BADL), Instrumental Activities of Daily Living (IADL).

**Table 2 diagnostics-13-00909-t002:** Drugs used to manage hyperactive delirium (not alcohol-induced).

Potentially Useful Drugs	Drugs to Be Avoided
If 4AT > 3 and 0 < mRASS ^1^ < 3 then Trazodone low dose (e.g., 50 mg) and melatonin If 4AT > 3 and mRASS ≥ 3 then Trazodone IV or IM (e.g., 25 mg IV bid) Or Haloperidol IM 0.5 to 1 mg up to a maximum dose of 5 mg per day as needed	All benzodiapines. High anticholinergic properties ^2^: Amitriptyline Chlorpromazine Clomipramine Clozapine Olanzapine Perphenazine Promazine Promethazine Quetiapine Thioridazine Trifluoperazine Trimipramine
The above suggestions are suggested for the management of hyperactive delirium in the absence of definite evidence of efficacy. Treating the causes of delirium is the only effective way to resolve delirium and can require the intervention of a geriatrician.	

^1^ The modified Richmond Agitation Sedation Scale (mRASS) is a tool to measure a patient’s level of agitation or sedation. A score of 1 stands for restless, 2 for agitated, 3 for very agitated (pulls or removes tubes or catheters, aggressive), and 4 for combative (overtly combative, violent, danger to staff). ^2^ Here we mentioned only the most used (for further information, https://intercheckweb.marionegri.it, accessed on 30 December 2022).

## Data Availability

Not applicable.

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
