# Peer review of "Asymptomatic Aortic Stenosis in an Older Patient: How the Geriatric Approach Can Make a Difference"

_diagnostics, 2023, doi:10.3390/diagnostics13050909_

Round 1
Reviewer 1 Report
Given the ageing of the population, aortic stenosis (AS) is becoming a frequent disease encountered in clinical practice. Thanks to the progress in this field, more and more people may be offered therapeutic options and, more often, at an older age. In this case report, some crucial points about the patient with AS who is referred for treatment are described. In particular, the lack of a patient-centered approach, in favor of a pathology-focused one, may be the responsible of the underestimation of potential problems in the management of the patient. I think that the geriatrician’s assessment should be included in the Heart Team evaluation, especially in case of the elderly population.
This topic is current and interesting, although I have some clarifications to ask.
1) In the case record, there is no mention to any kind of evaluation of the cognitive status of the patients before the discussion in the Heart Team, despite a mild cognitive impairment has been reported. In some Centers, a pre-operative assessment with MMSE is performed. Isn’t it routine to perform a preliminary cognitive evaluation in this Center? Was the reported CGA assessed after the procedure?
I think that one mention about how much the CGA assessment would have influenced the therapeutic choice may be worthwhile. In current literature, are there any recommendations for pre-treatment cognitive evaluation of elderly patients? Can you suggest some characteristics of the patient which make the geriatric evaluation highly recommended?
2) Although intuitive, the Authors should underline the main reasons for choosing TAVI based on the data described.
3) When reporting BNP, the reference values may be reported.
4) When reporting the echocardiography, I would indicate the grade of diastolic disfunction based on the international guidelines of the American Society of Echocardiography, instead of the single parameters.
5) Finally, the term ‘aortic stenosis’ is repeated in the text in full instead of abbreviated, as well as ‘BNP’.
Reviewer 2 Report
In the current state, the manuscript holds numerous language mistakes that need to be corrected before a potential publication.
While the issue of frailty among geriatric patients and including geriatrician among members of the heart team represents an important idea for personalized and modern approach towards medicine, which would optimize patient care, cost, and plan advanced multidisciplinary care, the authors present a case report that is supportive of the necessity of geriatric intensive care units rather than inclusion of a geriatrician among members of a heart team. A more representable example of a collaboration with a geriatrician would be mandatory to support the need of modification of the heart team, since infections are known to decompensate geriatric patients independently of assigned invasive treatment.
